# A Compilation Based Approach to Finding Centroids and Minimum Covering States in Planning

**Erez Karpas**

Technion – Israel Institute of Technology, Haifa, Israel
karpase@technion.ac.il

## Abstract

In some scenarios, an agent may want to prepare for achieving one of several possible goals, by reaching some state which is close (according to some metric) to all possible goals. Recently, this task was formulated as the finding centroids (which minimize the average distance to the goals) or minimum covering states (which minimize the maximum distance). In this paper, we present a compilation based approach for finding such states. Our compilation is very similar to the one used to find the worst case distinctiveness (wcd) in goal recognition design (GRD), and is orders of magnitude faster than the previous state-of-the-art, which was based on exhaustive search.

## Introduction

Automated planning typically deals with a scenario where a single agent is trying to achieve a single goal. However, in some cases an agent may want to prepare for achieving one of several possible goals, by reaching some state which is close (according to some metric) to all possible goals. Recently, this task was formulated (Pozanco et al. 2019), and two specific metrics were proposed – minimizing the average distance, and minimizing the maximum distance. The states which minimize these metrics are called centroids or minimum covering states, respectively.

Figure 1 illustrates such a setting. In this setting, the blue square is the initial state, and there are two possible goals: the red and green squares. To minimize the average distance to the possible goals (as well as the maximum distance), the agent should move up, following the gray arrow. From this state, the agent can follow the red arrow to the red square, or the green arrow to the green square.

Note that Figure 1 is very similar to the Figure illustrating worst case distinctiveness (wcd) in goal recognition design (GRD) (Keren, Gal, and Karpas 2019). Recall that the wcd in a goal recognition setting is the maximal number of actions an agent can take before an observer can know the exact goal the agent is aiming at, assuming the agent is optimal. In fact, the task of finding the wcd is very similar to the task we are concerned with here. Thus, in this paper, we present a compilation based approach for finding centroid and minimum covering states.

The compilations we present here are similar to the ones used for finding wcd (Keren, Gal, and Karpas 2014, 2015) in

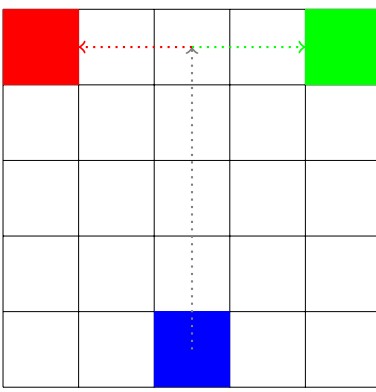

**Figure 1:** Illustration of Centroid.
The blue square represents the initial position of the agent, and the red and green squares represent the two possible goals. The gray arrow represents the path to a centroid state, while the red and green arrows represent the paths from the centroid state to each of the goals. Note that any square on the top row has an average goal distance of 2, and thus they are all centroid states, but only the middle square of the top row is a minimum covering state, with a maximum goal distance of 2.

that they model a set of agents performing actions together until some point, and then splitting – each pursuing its own goal. The main difference is in action costs: wcd attempts to maximize the shared portion of the plans, and thus the actions before splitting get a slight discount. In our task of interest here, we only care about the cost incurred after splitting, so the actions performed together become 0-cost actions. Our empirical evaluation shows that this compilation based approach is orders of magnitude faster than the previous approach (Pozanco et al. 2019).

## Preliminaries

The setting we consider here, which is the same as the setting in the previous work (Pozanco et al. 2019), is similar to STRIPS (Fikes and Nilsson 1971), except that there are multiple possible goals. Formally, $\Pi = \langle F, A, I, \mathcal{G}, C \rangle$, where:

- $F$ is a set of facts describing the possible states of the world, $2^F$.

- $A$ is a set of actions – each action $a \in A$ consists of a

set of preconditions $pre(a)$, add effects $add(a)$ and delete effects $del(a)$. Applying $a$ is possible in a state $s$ where $pre(a) \subseteq s$, and results in the state $s[\![\langle a \rangle]\!] = (s \setminus del(a)) \cup add(a)$. The cost of action $a$ is $C(a)$.

- $I \subseteq F$ is the initial state of the world, and

- $\mathcal{G}$ is a set of possible goals, where each possible goal $G \in \mathcal{G}$ is a set of facts $G \subseteq F$. A state $s$ satisfies a goal if $G \subseteq s$.

A path $\pi$ is a sequence of actions. A path $\pi = \langle a_0, a_1, \ldots a_n \rangle$ is applicable from state $s_0$ if $a_0$ is applicable at $s_0$ and $\langle a_1, \ldots a_n \rangle$ is applicable at $s_1 := s_0[\![\langle a_0 \rangle]\!]$. We denote the state reached by following path $\pi$ from state $s$ by $s[\![\pi]\!]$.

The cost of a path $\pi = \langle a_0, a_1, \ldots a_n \rangle$ is the sum of action costs $C(\pi) = \sum_{i=0}^{n} C(a_i)$. The optimal cost to go from some state $s$ to some set of facts (e.g., a goal) $G$, denoted $h^*(s, G)$ is the cost of a cheapest path from $s$ to some state $s_G$ which satisfies $G$, that is $\min_{\pi | G \subseteq s[\![\pi]\!]} C(\pi)$.

We can now define centroids and minimum covering states. In the presentation here, we reformulate the definitions from the previous work (Pozanco et al. 2019) to make them easier to follow. The average cost from some state $s$ to the possible goals $\mathcal{G}$ is $\frac{1}{|\mathcal{G}|} \sum_{G \in \mathcal{G}} h^*(s, G)$. A state $s$ is a centroid state if (a) it is reachable from the initial state, and (b) it minimizes the average cost to the possible goals. Similarly, the maximum cost from some state $s$ to the possible goals is $\max_{G \in \mathcal{G}} h^*(s, G)$, and a state $s$ is a minimum covering state if it is reachable from the initial state and minimizes the maximum distance to the possible goals.

To find optimal centroids or minimum covering states, previous work (Pozanco et al. 2019) proposed an exhaustive search approach which searches through all reachable states. From each state, this approach computes the cost of an optimal path to each possible goal by calling an optimal planner. The optimal centroid or minimum covering states is then found by looking for the best metric among all states generated by the search. Note that, with this approach, no pruning can be done on the states without losing optimality – unlike a regular $A^*$ search which does not expand nodes whose $f$-value is greater than the cost of the optimal solution.

## The Compilation

In this paper, we present a compilation based approach to finding centroids and minimum covering states. We begin by presenting a compilation for finding centroids.

## Finding Centroids

We present the compilation for finding centroids, which is very similar to the *latest-split* compilation (Keren, Gal, and Karpas 2014, 2019) for finding the worst case distinctiveness (wcd) in goal recognition design (GRD). As a reminder, in a similar setting to the one above, the worst case distinctiveness is the maximal number of actions an agent can take before an observer can know the exact goal the agent is aiming at, assuming the agent is optimal.

The *latest-split* compilation finds the wcd by optimally solving a classical planning task, where agents can either perform actions together, or split and perform actions separately – noting that after the agents split, they cannot perform actions together anymore. In finding the wcd, the objective is to find the longest possible sequence of actions before the agents split (i.e., the wcd). Thus, separate actions cost the same as they would normally, while actions performed together get some small discount $\epsilon$.

The only difference between the compilation we present here and the *latest-split* compilation is in the action costs. Specifically, we can think of performing actions together as going to the centroid state together, and the separate actions as the actions going from the centroid to each goal. Thus, we only care about the costs of actions splitting, so the separate actions cost the same as they normally would, while actions performed together cost 0. We now present the compilation formally.

Let $\Pi = \langle F, A, I, \mathcal{G} = \{G_1, \ldots G_n\}, C \rangle$ be a planning setting with multiple possible goals. Then the centroid compilation yields the classical planning task $\Pi' = \langle F', A', I', G', C' \rangle$, where:

- $F' = \{f_i \mid f \in F, i = 1 \ldots n\} \cup \{\text{split}, \text{unsplit}\}$, that is $f_i$ is a copy of fact $f$ for goal $G_i$.

- $A' = \{a_i \mid a \in A, i = 1 \ldots n\} \cup \{a_t \mid a \in A\} \cup \{\text{do-split}\}$, where

  - $a_i$ is the separate version of action $a$ for goal $i$, with $pre(a_i) = \{f_i \mid f \in pre(a)\} \cup \{\text{split}\}$, $add(a_i) = \{f_i \mid f \in add(a)\}$, $del(a_i) = \{f_i \mid f \in del(a)\}$, and $C(a_i) = C(a)$. That is, $a_i$ affects only the copy of the state for goal $G_i$, and is only possible after splitting.

  - $a_t$ is the together version of action $a$, with $pre(a_t) = \{f_j \mid f \in pre(a), j = 1 \ldots n\} \cup \{\text{unsplit}\}$, $add(a_t) = \{f_j \mid f \in add(a), j = 1 \ldots n\}$, $del(a_t) = \{f_j \mid f \in del(a), j = 1 \ldots n\}$, and $C(a_t) = 0$. That is, $a_t$ affects all copies of the state, and costs 0, and is possible only before splitting.

  - The do-split action allows the agents to split. $pre(\text{do-split}) = \{\text{unsplit}\}$, $add(\text{do-split}) = \{\text{split}\}$, $del(\text{do-split}) = \{\text{unsplit}\}$, and $C(\text{do-split}) = 0$.

- $I' = \{f_i \mid f \in I, i = 1 \ldots n\} \cup \{\text{unsplit}\}$, that is, the initial state is duplicated among all copies for all possible goals, with an indication that the agents have not split yet.

- $G' = \{f_i \mid f \in G_i, i = 1 \ldots n\}$, that is, the goal for agent $i$ is $G_i$.

We now prove that an optimal solution for $\Pi'$ gives us a centroid state for the original task $\Pi$.

**Theorem 1.** *An optimal solution for $\Pi'$ gives us a centroid state for the original task $\Pi$.*

*Proof.* Let $\pi'$ be any solution for $\Pi'$. Denote by $\pi'_i$ the sequence of actions in $\pi'$ that affects agent (goal) $i$, that is, the subsequence of $\pi'$ consisting of either $a_t$ or $a_i$ actions. It is easy to see that $\pi'_i$ encodes a path leading from $I$ to $G_i$, as (a) $\pi'_i$ contains exactly the set of actions affecting the $f_i$ facts, and (b) the goal for the $f_i$ facts is $G_i$. Thus, $\pi'$ encodes a set of paths leading from $I$ to each possible goal $G_i \in \mathcal{G}$.

Let us now consider a solution $\pi'$ for $\Pi'$, which does not contain the do-split action. This solution cannot apply any $a_i$ actions, as they require split, which is only achieved by the do-split action. Thus, it must have achieved all the goals together by $a_t$ actions, meaning that $\pi'$ reached some state $s$ which satisfies all possible goals – $s$ is a centroid, since the cost from $s$ to all possible goals is 0.

Now assume $\pi'$ does contain the do-split action. Denote by $s_{\pi'}$ the state in which the do-split action is performed, and by $\pi'_t$ the subsequence of $a_t$ actions performed to reach $s_{\pi'}$. It is easy to see that the cost of $\pi'_t$ in $\Pi'$ is 0 (as the $a_t$ actions cost 0). Now denote the remainder of $\pi'_i$ after following $\pi'_t$ by $\pi''^r_i$ – that is $\pi'_i = \pi'_t \cdot \pi''^r_i$ (this is possible since the $a_t$ actions are shared between all possible goals). Then $C(\pi') = \sum_{i=1}^{n} C(\pi''^r_i)$, since (a) all other actions cost 0, and (b) no action is shared between different goals after splitting. Thus, the cost of $\pi'$ is the sum of costs of reaching all possible goals from state $s_{\pi'}$.

Now consider an optimal solution $\pi'$ of $\Pi'$. Since the number of goals is a constant, minimizing the sum of costs is the same as minimizing the average cost. Thus, an optimal solution minimizes $\frac{1}{|\mathcal{G}|} \sum_{G \in \mathcal{G}} h^*(s_{\pi'}, G)$ – that is, finds a centroid. $\qquad\square$

Having shown that this compilation is correct, we now describe some simple optimizations for reducing the branching factor of the resulting search space. These optimizations have been presented before in the *latest-split* compilation.

Specifically, we aim to reduce the branching factor after splitting, which increases by a factor of $n$ (since we can now apply $a_i$ for $i = 1 \ldots n$ instead of just $a_t$). To address this, we fix an order between the agents, and only allow the compilation to pursue goal $i + 1$ after goal $i$ has been achieved.

This is done by adding $n$ new facts $\{\text{done}_i \mid i = 1 \ldots n\}$. $\text{done}_{i-1}$ is added to the preconditions of all $a_i$ actions, for $i = 2 \ldots n$ ($a_1$ actions remain the same). Additionally, we add $n$ new actions called $\text{end}_i$ actions for $i = 1 \ldots n$, with $pre(\text{end}_i) = G_i, add(\text{end}_i) = \{\text{done}_i\}$, and $add(\text{end}_i) = \emptyset$, and $C(\text{end}_i) = 0$ – these actions mark that goal $G_i$ has been achieved, and the compilation can move on to achieving $G_{i+1}$.

This optimization does not change the proof of correctness, since every solution to the compilation still encodes $n$ solutions to the $n$ possible goals, and the cost of a solution is still the sum of actions costs after splitting. However, the branching factor is reduced considerably.

## Finding Minimum Covering States

We now move on to presenting the compilation for finding minimum covering states. Unfortunately, unlike the average (or sum) that is used in centroids, the max operator in minimum covering states is not additive. Thus, we do not present a compilation which directly finds a minimum covering state.

Instead, we present a compilation which, given some cost budget $B$, checks whether there is some reachable state $s$ such that the maximum cost of reaching any possible goal $G_i \in \mathcal{G}$ is at most $B$ – that is, whether $\max_{G \in \mathcal{G}} h^*(s, G) \leq$

$B$. By performing a binary search over $B$ it is possible to find the exact minimum covering value (and state).

The first compilation we present uses numerical variables to keep track of the budget spent to reach each goal. Specifically, the compilation is the same as the above compilation for finding centroids, except that we add $n$ new numerical variables, $B_1 \ldots B_n$. The value of $B_i$ in the initial state is 0. We also modify the $a_i$ actions, and add $B_i < B - C(a_i)$ to $pre(a_i)$, and $B_i + = C(a_i)$ to the effects of $a_i$ – that is, action $a_i$ keeps track of the budget spent to reach $G_i$, and makes sure this budget does not go over $B$. $a_t$ actions do not modify the variables, since we only care about the cost of reaching the goals from a minimum covering state (after splitting).

Note that although numerical planning is undecidable in general, this compilation is a special case of numerical planning where numerical variables are only compared to constant, and are only increased by a constant. Therefore, the numerical planning task described here is decidable (Helmert 2002).

**Theorem 2.** *Let $\Pi'$ be a numerical planning task with budget $B$ as described above. Then $\Pi'$ is solvable iff there exists some reachable state $s$ such that $\max_{G \in \mathcal{G}} h^*(s, G) \leq B$.*

*Proof.* As in Theorem 1, it is easy to see that any solution $\pi'$ yields a state $s_{\pi'}$ in which the do-split action was performed, unless there is a state which satisfies all the goals, in which case the minimum covering distance is 0.

Define $\pi'_i, \pi'_t$, and $\pi''^r_i$ as in Theorem 1, such that $\pi'_i = \pi'_t \cdot \pi''^r_i$. It is easy to see that $C(\pi''^r_i) \leq B$, as these are the only actions which modify $B_i$, and their preconditions enforce that the cost never increases past $B$. Thus $\max_{G \in \mathcal{G}} h^*(s_{\pi'}, G) \leq B$. $\qquad\square$

Having discussed the general case, we now present a compilation to classical planning which is able to find the minimum covering state directly – for the case when all actions have unit cost. This compilation is similar to the *sync-latest-split* compilation for finding the wcd with non-optimal agents, when agents have a deception budget (Keren, Gal, and Karpas 2015).

The compilation extends the basic compilation for finding centroids, without the optimization for enforcing the order between the agents. In this compilation, after splitting agents take turns executing actions in a round robin manner. This is implemented by adding $n$ new facts, $\text{turn}_i$ for $i = 1 \ldots n$. For each $a_i$ action, we add $\text{turn}_i$ to $pre(a_i)$, $\text{turn}_{i+1 \bmod n}$ to $add(a_i)$, and $\text{turn}_i$ to $del(a_i)$, to keep track of whose turn is next.

This turn taking mechanism ensures that when all agents have reached their goals, they have all executed almost the same number of actions – up to a difference of 1 because some agents may have acted in the last and other may have not. To account for this while keeping track of the minimum covering cost, only the actions of agent 1 have a cost (of 1, since all actions are unit cost). All other actions ($a_t$ actions and $a_i$ actions for $i > 1$) cost 0. This ensures that the compilation only counts the costs incurred by the first agent, who is always the first to reach the higher number of steps.

| | E | C |
|---|---|---|
| BLOCKS-WORDS | | |
| 1 | 838.04 | 25.89 |
| 2 | 835.30 | 16.05 |
| 3 | 852.02 | 11.03 |
| 4 | 821.71 | 22.96 |
| 5 | 818.97 | 7.18 |
| 6 | 835.56 | 15.58 |
| 7 | 810.36 | 9.06 |
| 8 | 818.74 | 8.69 |
| 9 | 827.43 | 16.09 |
| 10 | 830.51 | 12.18 |
| AVG | 828.86 | 14.47 |
| RANGER | | |
| 1 | 2197.42 | 14.31 |
| 2 | 2102.58 | 16.46 |
| 3 | 3124.19 | 17.16 |
| 4 | 1984.45 | 14.23 |
| 5 | 2140.93 | 16.59 |
| 6 | 1974.24 | 14.57 |
| 7 | 2126.50 | 17.13 |
| 8 | 2227.33 | 16.02 |
| 9 | 2128.61 | 14.31 |
| 10 | 2371.44 | 15.95 |
| AVG | 2237.77 | 15.67 |

**Table 1:** Search Time for Finding Centroids

| | E | Cd | Cb |
|---|---|---|---|
| BLOCKS-WORDS | | | |
| 1 | 824.75 | 136.61 | 301.94 |
| 2 | 867.98 | 434.46 | 484.27 |
| 3 | 853.32 | 198.46 | 311.41 |
| 4 | 812.95 | 31.96 | 214.00 |
| 5 | 814.90 | 32.21 | 257.64 |
| 6 | 826.78 | 403.20 | 538.93 |
| 7 | 833.61 | 44.56 | 282.84 |
| 8 | 805.54 | 84.80 | 352.90 |
| 9 | 820.50 | 27.13 | 207.28 |
| 10 | 827.93 | 97.37 | 317.67 |
| AVG | 828.83 | 149.07 | 326.89 |
| RANGER | | | |
| 1 | 2162.87 | TO | TO |
| 2 | 2118.05 | TO | TO |
| 3 | 2732.27 | TO | TO |
| 4 | 2042.48 | TO | TO |
| 5 | 2189.26 | TO | TO |
| 6 | 2031.07 | TO | TO |
| 7 | 2155.46 | TO | TO |
| 8 | 2169.37 | TO | TO |
| 9 | 2199.38 | TO | TO |
| 10 | 2368.51 | TO | TO |
| AVG | 2216.87 | - | - |

**Table 2:** Search Time for Finding Minimum Covering States

Finally, another potential issue is that some agent might reach its goal, and then be forced to continue acting due to the turn taking mechanism. To eliminate this issue, we also introduce a set of NOOP actions – one for each agent. $pre(\text{NOOP}_i) = G_i$, and $add(\text{NOOP}) = del(\text{NOOP}) = 0$, with costs $C(\text{NOOP}_1) = 1$ and $C(\text{NOOP}_i) = 0$ for $i > 1$. These NOOP actions allow agent $i$ to stay at its goal once it reaches it. Of course, the NOOP actions also implement the turn taking mechanism described above and can only be executed after splitting, but we omit these from the description for the sake of clarity.

As before, it is easy to see that any solution to this new compilation encodes $n$ different solutions, one for each goal. The only actions that incur any cost are the actions of agent 1 after splitting, who is always the agent who has executed the most actions. Thus, the cost of any solution is the maximal cost of reaching any goal after splitting, and an optimal plan finds a minimum covering state.

## Empirical Evaluation

We implemented our compilation in Python[1], and compare it to the exhaustive search approach (Pozanco et al. 2019)[2]. In both cases, we use the same planner used in the exhaustive search approach – the Fast Downward (Helmert 2006) planner with the A* search algorithm (Hart, Nilsson, and Raphael 1968) and the lmcut heuristic (Helmert and Domsh-

---

[1]Available at https://github.com/karpase/grs_compilation

[2]We used the implementation available at https://github.com/apozanco/GRS_0.1

lak 2009). In our case we use it to optimally solve the compilation, and in the exhaustive search approach to find the costs of optimal plans.

We used the same time and memory limits as the exhaustive approach – 3600 seconds and 16GB of memory – running on a server with a Xeon E5-2695 CPU. We compared both approaches on the same domains as the exhaustive approach, obtained from their software repository: BLOCKS-WORDS and RANGER. The BLOCKS-WORDS domain involves 5 blocks with different letters on them, where possible goals describe a word to be spelled with these blocks (there are 3 possible goals in each problem). The RANGER domain involves navigating in a 20x20 grid, where 20% of the cells are blocked, where possible goals described a specific location for the agent to reach (there are 4 possible goals in each problem).

Table 1 shows the search time for finding centroids, comparing the exhaustive approach (E) to our compilation (C). These results show a clear advantage for our compilation based approach, which is about 2 orders of magnitude faster than the exhaustive search approach.

Table 2 shows the search time for finding minimum covering states, comparing the exhaustive approach (E) to our direct compilation (Cd) and to binary search using our compilation (Cb). Although the binary search is defined for numerical planning we converted it to classical planning by converting numerical variables to discrete ones (which is possible only when actions have uniform cost). This allows us to compare all 3 approaches using the same underlying planner, even though the binary search approach can use a satisficing planner instead. These results show a clear advantage for the compilation based approach in BLOCKS-WORDS. However, for the RANGER domain, both of our compilations time out (at 3600 seconds), while the exhaustive search approach needs about 60% of that time to solve these problems. We believe this is because the compilation for finding minimum covering states has many 0 cost actions (as the planner only pays for increasing the budget), leading to many 0-cost plateaus.

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
