# OpenReview forum: "A Compilation Based Approach to Finding Centroids and Minimum Covering States in Planning"
_icaps-conference.org/ICAPS/2021/Workshop/HSDIP — HSDIP 2021_

### Official Review · AnonReviewer2 · 2021-05-21

**Confidence:** 4
**Overall Score:** Accept

**Review:**

The paper presents a novel approach to determine centroids and minimum covering
states in classical planning based on a compilation. This compilation approach
has its origin in the field of goal recognition, where a compilation was
introduced for the worst-case distinctivity (WCD), i.e. the maximum cost of a
path that an agent can follow before its goal can be inferred. Small changes to
this compilation are introduced and it is explained why by doing so centroids
of a given problem can be computed. The correctness of this compilation is
proved. Furthermore, a compilation for determining minimum covering states is
presented in a similar way, but based on a bounded planning approach. Finally,
an empirical evaluation is conducted, which shows that the compilation approach
performs favourably compared to the previous state of the art, which
is based on exhaustive search.

Although the compilations presented are quite similar to those presented for
WCD (Keren, Gal and Karpas 2019), it is a contribution in itself to show that
these can be used, with some modifications, to compute centroids and minimal
covering states. The intuition of why and how this is possible is well
described in the paper. The empirical results are also convincing. Overall, I
have a few comments mainly on the presentation, which I list below.

With respect to Figure 1, my understanding is that there is a single minimum
coverage state, namely the middle cell of the top row with a (minimum) distance
to both goals of 2. However, my understanding is that there are 5 centroid
states, namely all cells in the top row with an average distance of 2 each. Is
this correct? It would be nice to clarify this and explain it in the text.

In the compilation, F' is a bit ambiguous. If I understand it correctly, each
fact is copied n times, with the i-th copy corresponding to goal i:  Probably
something like {f_i | i = 1...n, f \in F).

I would recommend giving a brief explanation of what WCD is in the
introduction, as is done in section Finding Centroids ("As a reminder, ...").
As someone who is not fully familiar with the literature on goal recognition,
not giving this explanation at the beginning of the paper made it a bit more
complicated than necessary.

The paper presents an interesting and empirically well-performing compilation
approach to finding centroids and minimum covering states in classical
planning. Overall, I think it is a valuable addition to the HSDIP workshop,
which is why I recommend accepting it.

Minors:
- Last sentence of the introduction does not seem to be complete.
- Preliminaries: ... is similar to *sc* strips … => ... is similar to strips … ?
- Preliminaries: Formally, \pi = … => Formally, \Pi …
- Preliminaries: A path is defined as \pi, which is not really common as it is usually
used for a plan. I would consider referring to it as \pi with an arrow on top
(\overrightarrow{\pi}).
- Proof of Theorem 1: … affect *agent* i, … => ... affect goal i ... ?

---

### Official Review · AnonReviewer1 · 2021-05-26

**Confidence:** 4
**Overall Score:** Accept

**Review:**

This paper introduces a new compilation from the problems of finding centroid and maximum covering states to classical planning.

This is an interesting idea, likely to be of interest of the HSDIP audience. The paper does not focuses too specifically on heuristics or search algorithms but still is related as these algorithms are ultimately used within the approach (I think perhaps some changes could be done to improve this, see comments below).

The paper is well written, the compilation is well explained and motivated. My only comments about the paper clarity are:

  - in the Preliminaries section that mentions that the definitions are slightly different than the ones in previous work. This is very unclear, specially since a direct comparison against previous work is presented in the empirical results. What are exactly the differences? Are the algorithms in the evaluation computing the same centroids and minimum covering states?

  - In several occasions, it is mentioned that the previous work performed an "exhaustive" search. It is not complete clear how this differs from the proposed approach. Solving a classical planning task optimally also requires some form of exhaustive search, doesn't it? I think the difference between performing heuristic search (A* with lmcut) on the compiled task, and the previous exhaustive search approach could be better explained. Also, this would strengthen the relationship with the main topic of the workshop.


The experimental results are convincing, outperforming the previous approaches. However, the evaluation could also be strengthened in several ways:

 - From the point of view of HSDIP, perhaps it would be interesting to include a comparison of different planners and heuristics on the new instances. While it is true that LM-cut is convenient and allows a more direct comparison with previous work, but this could be complemented with a more broad analysis of heuristics on the compiled instances.

 - All the instances used have very similar runtimes. This suggests that they have a similar size and number of goals. The paper could be significantly stronger if the comparison separately analyzed how the different approaches scale with respect to the two dimensions: problem size and number of goals.

 - The evaluation could be extended to more domains. I know that these were used by Pozanco et al. However, this evaluation was extended for his thesis (https://apozanco.github.io/thesis_apozanco.pdf) so the benchmarks are likely to be available upon request.


  Minor comments:
 - page 1, the introduction ends  on a half sentence
 - page 2, "We begin by " is repeated twice in a row. I'd remove the sentence before the "Finding Centroids" subsection
 - page 2, Definition of F' is missing f \in F
 - page 2, which does not contains -> contain
 - page 2, can not -> cannot
 - page 3, Proof of thm 1 ends on "," instead of .
 - page 3, Thm 2, there some -> there exists or there is some
 - page 3, NOOP) = 1) and NOOP) = 0) -> there are two extra closing parenthesis
 - page 4, approached -> approaches

---

### Author Response · Authors · 2021-05-30
**Review Response**

We would like to thank the reviewers for their helpful feedback. All comments will be addressed in the final version, but we respond to the major ones here.


R1:
* The definitions of centroids and minimum covering states define the same concept, we just reformulated the definitions to make them clearer - this is now clarified in the text.

* The exhaustive search in the previous work had to search all reachable states. This is in contrast to A*, which does not expand nodes where f(s) > C*. We have added a paragraph explaining this at the end of the preliminaries section.

* The previous approach uses FD with A*/lmcut as its underlying optimal planner, which is why we chose this. To compare using different planner, we would need to also replace the underlying optimal planner there - we will do this for a full conference paper next year.

* Indeed, we used the existing benchmarks where blocks-words instances all have 5 blocks, and ranger instances are 20x20 grids with 20% obstacles. We will explore scaling behavior for a full conference paper next year.

* Thanks for the suggestion to include more domains and thr pointer. We will do this for a full conference paper next year.

We also fixed all the minor issues, thank you.

R2:

* You are correct about Figure 1, and we have added this to the caption of the figure.

* You are also correct about F', and we fixed this.

* We added an explanation about WCD in the intro.

We also fixed all the minor issues, thank you.

---

### Decision · Program_Chairs · 2021-06-10

**Decision:**

Accept

**Comment:**

There is a clear consensus that this paper is well written and should be accepted. Congratulations. Please make sure to include the (minor) points that come up in the reviews.